# A systematic review protocol: Efficacy and safety of nitrous oxide in analgesia in burn patients with dressing change

Weifeng Wang[1,2]☯, Xianli Meng[3]☯, Yan Zhao[4]☯, Wei Gong[5]☯, Xiaochen Jiang[1,2], Wenjuan Cao[1,2], Xueling Qiu[2,6], Chenxi Sun[2,7], Fan Sun[2,8], Yuchen Wang[2,7], Lu Tang [2]*

1 School of Nursing, Shandong Second Medical University, Weifang, China, 2 Department of Stomatology, the 960th hospital of People's Liberation Army of China (PLA), Jinan, China, 3 Changhai Hospital, Naval Medical University, Shanghai, China, 4 Dalian Rehabilitation Center of the Joint Logistics Support Force of the Chinese People's Liberation Army, Dalian, China, 5 Department of the Food and Drug Inspection, Drug and Instrument Supervision and Inspection Station of Shenyang Joint Logistics Support Center, Shenyang, China, 6 School of Nursing, Shandong First Medical University, Taian, China, 7 School of Nursing, Jinzhou Medical University, Jinzhou, China, 8 School of nursing, Shandong University of Traditional Chinese Medicine (TCM), Jinan, China

☯ These authors contributed equally to this work.
* tanglu_office@163.com

## Abstract

### Background

To alleviate pain in burn patients during dressing changes, it is necessary to identify an effective analgesic method. Conventional opioid analgesics have many limitations. Nitrous oxide is a fast-acting, safe and reversible inhaled analgesic gas. This systematic review will evaluate the effectiveness and safety of nitrous oxide in the treatment of pain during dressing changes in burn patients.

### Method

The protocol was developed according to the PRISMA-P checklist and registered on PROSPERO (CRD42024550197). A systematic search will be performed in the following databases: PubMed, EMBASE, Web of Science, Cochrane Library to identify clinical trials comparing nitrous oxide inhalation with standard care in pain management during dressing changes in burn wounds. The search of all databases will be conducted on October 15, 2025.Our search scope will include studies published between each database creation and search date.Two researchers will independently screen studies, extract data, and evaluate study quality using the Risk of Bias2 tool. Primary outcomes will include pain, anxiety, side effects, among others.R statistical software (version 4.3.1) and R studio will be used to perform meta-analyses.Effect size will be expressed by 95% confidence interval (CI) of weighted mean difference (MD) and risk ratio (RR). Subgroup analyses and sensitivity analyses will be performed to explore sources of heterogeneity and assess the robustness of the

**Data availability statement:** This is a systematic review study protocol and will not generate new data. All relevant data for this study are available from the corresponding author upon completion of the study.

**Funding:** The author(s) received no specific funding for this work.

**Competing interests:** The authors have declared that no competing interests exist.

results.Publication bias will be assessed using funnel plot and Egger test. We will use the Grading of Recommendation, Evaluation, Development and Evaluation (GRADE) to assess the quality of the evidence.

## Discussion

Operative pain has always been a difficult problem for burn patients. This study will evaluate the analgesic effect of nitrous oxide on dressing change in burn patients through comprehensive search and rigorous methods, and provide evidence support for clinical decision-making.

## Introduction

Pain control in burn patients has always been a challenge. Effective pain management can significantly reduce suffering for burn patients and prevent neuropathic pain and chronic pain [1].

Burn debridement and dressing changes can lead to great pain in patients. Extremely severe pain may cause serious negative emotions and even psychological problems for patients [2]. Uncontrolled moderate to severe pain can hinder wound care and physical therapy, and directly affect the speed and quality of burn wound healing. Therefore, appropriate analgesic measures are essential [3]. The application of analgesic drugs for burn prevention should consider not only the burn site, burn depth, burn area and debridement time but also changes in the pharmacodynamics and pharmacy [4,5]. Drugs should have the characteristics of quick onset, short action time, fewer active metabolites, and fewer adverse reactions [6].

Medications are usually used to reduce pain in burn patients. Even without surgery, burn patients can suffer greatly. The American Burn Association (ABA) guidelines state that opioids are considered the primary method of controlling burn pain [7]. Nonsteroidal anti-inflammatory drugs (NSAIDs), such as gabapentin, lidocaine, and ketamine, can also be used to treat pain in burn patients. However, long-term use of these drugs, especially opioids, may cause some compilations. These side effects include dizziness, nausea, respiratory depression and addiction [8]. Burn patients receive medication for pain relief every other day or even every day, so the probability of side effects is greatly increased [9].

Nitrous oxide is a colorless and odorless gas that has been used in the field of analgesia for 150 years and is still widely used in conscious analgesia [10]. Nitrous oxide has been shown to be a safe and fast-acting analgesic gas, with rapid induction and rehabilitation, mild adverse reactions, a low incidence of side effects, simple manipulation, and satisfactory analgesic effects in most cases [11–13]. However, some studies suggest that nitrous oxide has limited analgesic effects [14]. In addition, nitrous oxide also has sedative and antianxiety effects and is often used in combination with oxygen at a specific ratio for conscious anesthesia and analgesia outside the operating room [15], such as dental surgery sedation and analgesia [16], acute injury sedation and analgesia [17], and labor analgesia [18].

Nitrous oxide may be administered by a trained nurse who is not specialized in anesthesia. Previous studies on pain management mostly focused on the research of narcotic analgesics administered by anesthesiologists or clinicians, but few methods that can be administered by nurses according to doctors' advice without the presence of anesthesiologists.

At present, there is no strong evidence to analyze the effect of nitrous oxide in burn debridement and dressing changes. We will conduct a systematic review and meta-analysis of published clinical trials to evaluate the analgesic effectiveness and adverse effects of nitrous oxide compared with placebo or other measures during debridement and dressing change in burn patients. These findings will inform evidence-based clinical practice on pain management for debridement and dressing changes in burn patients.

## Materials and methods

The protocol was developed from the Preferred Reporting Items for Systematic Review and Meta-Analysis Protocols (PRISMA-P) declaration checklist [19]. The protocol has been registered in the PROSPERO database and assigned an identifier: CRD42024550197. This review is based on published studies, so the study design, process and results do not require patients or public participation or ethical approval.

### Search strategy

To complete this study, we will perform searches in the following databases: PubMed, EMBASE, the Cochrane library and the Web of Science. Completed but unpublished studies will also be included in this review. We will conduct a literature search using the following MeSH terms and text words related to nitrous oxide and burn debridement dressing changes: 'Nitrous Oxide'; 'Laughing Gas'; 'Burns'; 'Dressing'; 'Bandages'; 'Anesthesia and Analgesia'; 'Conscious Sedation'; 'Analgesia'; and 'Pain'.The search strategies are shown in Tables 1–4. We will supplement the literature by searching the following websites: the WHO International Clinical Trials Registry Platform and ClinicalTrials.gov. We will also conduct a manual search of gray literature (e.g., unpublished conference literature, reports) on sites such as Google Scholar, Open Access Theses and Dissertations.

No restrictions will be imposed on the study design, date or language.The search of all databases will be conducted on October 15, 2025, and all identified studies will be included in the title and abstract screening. Our search scope will include studies published between the creation of each database and the date of the search. We will perform a manual search of reference lists for prospective systematic reviews and meta-analyses to identify reports that may be relevant but omitted through electronic searches.

### Eligibility criteria

Studies that meet the following criteria will be included in this review.

**Study type.** We will consider randomized controlled trials(RCTs) that follow the PICOS framework, which structured the research question into its key components: P (Population), I (Intervention), C (Comparison), O (Outcomes), and S (Study design). A detailed explanation of each component is provided in the subsequent section.

**Participants.** The study was performed in burn patients and focused on debridement and dressing change.The subjects were patients of all ages who were hospitalized for the first time due to burns.This study mainly evaluates the effect of nitrous oxide on manipulative pain, therefore, there is no limit on dressing type.

**Intervention.** Inhalation of nitrous oxide during debridement and dressing change aims to reduce patient suffering.

**Comparators.** The control group received placebo or other analgesic methods. Approaches considered include: opioid analgesia, regional block analgesia,and non-pharmacological therapies for analgesia by diversion. Because these are common methods used clinically.

**Outcome measures.** Primary outcome for included trials must show the intensity of pain in the patient numerically, including the VAS(Visual Analogue Scale) score, NRS(Numerical Rating Scale) score, or FLACC(Face, Legs, Activity, Cry,

**Table 1. The search strategy for PubMed.**

| ORDER | STRATEGY |
| --- | --- |
| #1 | Search: "Nitrous Oxide"[Mesh] |
| #2 | Search: "laughing gas"[Title/Abstract] |
| #3 | Search: "dinitrogen oxide"[Title/Abstract] |
| #4 | Search: "N2O"[Title/Abstract] |
| #5 | Search: "Burns"[Mesh] |
| #6 | Search: "burn"[Title/Abstract] |
| #7 | Search: "burn injury"[Title/Abstract] |
| #8 | Search: "burn wound"[Title/Abstract] |
| #9 | Search: "scald"[Title/Abstract] |
| #10 | Search: "dressing change"[Title/Abstract] |
| #11 | Search: "wound dressing"[Title/Abstract] |
| #12 | Search: "bandage change"[Title/Abstract] |
| #13 | Search: "Analgesia"[Mesh] |
| #14 | Search: "pain relief"[Title/Abstract] |
| #15 | Search: "pain management"[Title/Abstract] |
| #16 | Search: "analgesic effect"[Title/Abstract] |
| #17 | Search: "Pain"[Mesh] |
| #18 | Search: "ache"[Title/Abstract] |
| #19 | Search: "soreness"[Title/Abstract] |
| #20 | Search: "discomfort"[Title/Abstract] |
| #21 | #1 OR #2 OR #3 OR #4 |
| #22 | #5 OR #6 OR #7 OR #8 OR #9 |
| #23 | #10 OR #11 OR #12 |
| #24 | #13 OR #14 OR #15 OR #16 |
| #25 | #17 OR #18 OR #19 OR #20 |
| #26 | #24 OR #25 |
| #27 | #21 AND #22 AND #23 AND #26 |

Consolability scale) score.Different burn severities and depths affect pain scores, so we used the change in pain scores before and after the intervention as the primary outcome measure. Different scales can also lead to heterogeneity, so we will use standardized mean differences for statistical analysis. The secondary outcomes included at least one of the following: level of anxiety, side effects, duration of surgery, or patient satisfaction.

## Screening

Two reviewers(WFW and XLQ) will independently screen for titles and abstracts selected from the search and identify studies based on inclusion and exclusion criteria. Reviewers (WFW and XLQ) then assess the eligibility of the full-text content to determine whether it is ultimately included. When multiple publications appeared in the same study population, the most recent report with the largest sample size and outcomes that met the eligibility criteria was selected. When there is a dispute, it is discussed by two reviewers to reach a consensus, and if necessary, with a third researcher(XCJ). For studies with non-English titles and abstracts, we will use machine translation tools such as Google Translate to translate them into English for initial evaluation. If there are still doubts after machine translation, the research will enter the full-text screening stage. For non-English full texts, we will use high-quality translation tools such as Google Translate or DeepL to generate an English translation of the full text as the basis for evaluation. In view of the advanced nature of today's

**Table 2. Search strategies for Embase.**

| ORDER | STRATEGY |
|---|---|
| #1 | 'nitrous oxide'/exp |
| #2 | 'laughing gas':ab,ti |
| #3 | 'dinitrogen oxide':ab,ti |
| #4 | 'N2O':ab,ti |
| #5 | 'burns'/exp |
| #6 | 'burn':ab,ti |
| #7 | 'burn injury':ab,ti |
| #8 | 'burn wound':ab,ti |
| #9 | 'scald':ab,ti |
| #10 | 'dressing change':ab,ti |
| #11 | 'wound dressing':ab,ti |
| #12 | 'bandage change':ab,ti |
| #13 | 'redressing':ab,ti |
| #14 | 'analgesia'/exp |
| #15 | 'pain relief':ab,ti |
| #16 | 'pain management':ab,ti |
| #17 | 'analgesic effect':ab,ti |
| #18 | 'pain'/exp |
| #19 | 'ache':ab,ti |
| #20 | 'soreness':ab,ti |
| #21 | 'discomfort':ab,ti |
| #22 | #1 OR #2 OR #3 OR #4 |
| #23 | #5 OR #6 OR #7 OR #8 OR #9 |
| #24 | #10 OR #11 OR #12 OR #13 |
| #25 | #14 OR #15 OR #16 OR #17 |
| #26 | #18 OR #19 OR #20 OR #21 |
| #27 | #25 OR #26 |
| #28 | #22 AND #23 AND #24 AND #27 |

translation tools, the probability of encountering ambiguous content is very low. The risk of bias assessment will be based on the translated text. In case of uncertainty, it will be noted in the text and taken into consideration in sensitivity analysis. The screening process and reasons for exclusion are shown in the PRISMA flowchart (Fig 1).

## Data extraction and records

The extraction of the data was performed independently by two reviewers(WFW and XLQ). Data that need to be extracted include: first author name, year of publication, year of baseline study, country, interventions, controls, blinding, subject race, number, sex, age, quantitative results,tool of measurement, narrative summary of findings(e.g., side effects). The results of all data are presented in tabular form. For studies with missing or incomplete data (e.g., missing mean or standard deviation), we attempted to contact the corresponding author by email. If no response is received or no data is provided. We will estimate missing means and standard deviations of available statistics (median and interquartile range) using validated methods. If data are always unavailable or calculated, we will exclude this study.We will prioritize extracting and pooling pain scores during the intervention. If the study reports multiple time points, we will treat each time point

**Table 3. Search strategies for Web of Science.**

| ORDER | STRATEGY |
|---|---|
| #1 | TS=("nitrous oxide") |
| #2 | TS=("laughing gas") |
| #3 | TS=("dinitrogen oxide") |
| #4 | TS=("n2o") |
| #5 | TS=(burns) |
| #6 | TS=(burn) |
| #7 | TS=("burn injury") |
| #8 | TS=("burn wound") |
| #9 | TS=(scald) |
| #10 | TS=("dressing change") |
| #11 | TS=("wound dressing") |
| #12 | TS=("bandage change") |
| #13 | TS=(analgesia) |
| #14 | TS=("pain relief") |
| #15 | TS=("pain management") |
| #16 | TS=("analgesic effect") |
| #17 | TS=(pain) |
| #18 | TS=(ache) |
| #19 | TS=(soreness) |
| #20 | TS=(discomfort) |
| #21 | #1 OR #2 OR #3 OR #4 |
| #22 | #5 OR #6 OR #7 OR #8 OR #9 |
| #23 | #10 OR #11 OR #12 |
| #24 | #13 OR #14 OR #15 OR #16 |
| #25 | #17 OR #18 OR #19 OR #20 |
| #26 | #24 OR #25 |
| #27 | #21 AND #22 AND #23 AND #26 |

as an independent analysis and perform subgroup analyses grouped by time to avoid repeated inclusion of data from the same group of patients in the same Meta-analysis. We do not make statistical adjustments for repeated measures.

### Subgroup analysis

If there is a sufficient number of subgroup samples, we will conduct subgroup analysis according to the following criteria: 1. Adults and children, because the FLACC scale used by children is mostly evaluated by medical staff through facial expressions and leg movements, while adults are mostly self-rated by VAS or NRS scale. 2. Different control group types, including placebo, opioid, regional block analgesia, and non-drug analgesia. 3. Different burn sites, burn depths or nitrous oxide concentrations. 4. The first or subsequent dressing changes will also affect the level of pain, so we will also conduct subgroup analysis based on the number of dressing changes.

**Assessment of risk of bias.** The study's risk of bias will be independently assessed by two researchers (WFW and XLQ) using the Cochrane Collaboration Risk of Bias 2 (RoB 2) tool. The RoB 2 tool includes five domains: bias arising from the randomization, bias due to deviation from established interventions, bias from missing outcome data, bias from outcome measures, and bias from selective reporting.Each domain will be classified as 'low risk' "some concern," or

**Table 4. Search strategies for Cochrane.**

| ORDER | STRATEGY |
|---|---|
| #1 | [mh "Nitrous Oxide"] |
| #2 | ("laughing gas"):ti,ab,kw |
| #3 | ("dinitrogen oxide"):ti,ab,kw |
| #4 | (N2O):ti,ab,kw |
| #5 | [mh Burns] |
| #6 | (burn):ti,ab,kw |
| #7 | ("burn injury"):ti,ab,kw |
| #8 | ("burn wound"):ti,ab,kw |
| #9 | (scald):ti,ab,kw |
| #10 | ("dressing change"):ti,ab,kw |
| #11 | ("wound dressing"):ti,ab,kw |
| #12 | ("bandage change"):ti,ab,kw |
| #13 | [mh Analgesia] |
| #14 | ("pain relief"):ti,ab,kw |
| #15 | ("pain management"):ti,ab,kw |
| #16 | ("analgesic effect"):ti,ab,kw |
| #17 | [mh Pain] |
| #18 | (ache):ti,ab,kw |
| #19 | (soreness):ti,ab,kw |
| #20 | (discomfort):ti,ab,kw |
| #21 | #1 OR #2 OR #3 OR #4 |
| #22 | #5 OR #6 OR #7 OR #8 OR #9 |
| #23 | #10 OR #11 OR #12 |
| #24 | #13 OR #14 OR #15 OR #16 |
| #25 | #17 OR #18 OR #19 OR #20 |
| #26 | #24 OR #25 |
| #27 | #21 AND #22 AND #23 AND #26 |

"high risk," and each trial overall risk of bias follows its highest risk of bias. The results of the quality assessment will be presented in tabular form, with each judgment accompanied by a brief justification [20].

## Data analysis

In this study, we will use R statistical software (version 4.3.1) and R studio. The meta-analysis will be conducted using the 'meta' package. The combined effect size of the quantitative data will be expressed by the weighted Standardized Mean Difference(SMD) with 95% confidence interval(CI) and dichotomous outcomes will be expressed by the risk ratio (RR) with 95% confidence interval (CI).Descriptive outcomes for which combined effect size cannot be calculated will be evaluate qualitatively. Q statistic ($p < 0.1$ indicates significance) and $I^2$ test will be used to analyze the heterogeneity of the included studies. $I^2$ values of 0–30%, 30%−50%, 50%−70% and 70–100% will be respectively considered low, moderate, considerable and substantial heterogeneity.If $p < 0.1$ and $I^2$ value is $< 50\%$, the included studies have low heterogeneity, using the fixed-effect model; If $p \leq 0.1$ and $I^2$ values are $\geq 50\%$, the heterogeneity of the included studies is high and the random effects model will be used. When the heterogeneity is large, the source of heterogeneity is explored through subgroup analysis. Potential heterogeneity will be explored by sensitivity analysis (sequentially excluding one study observing changes in pooled effect size), as well as the robustness of the outcome was assessed.

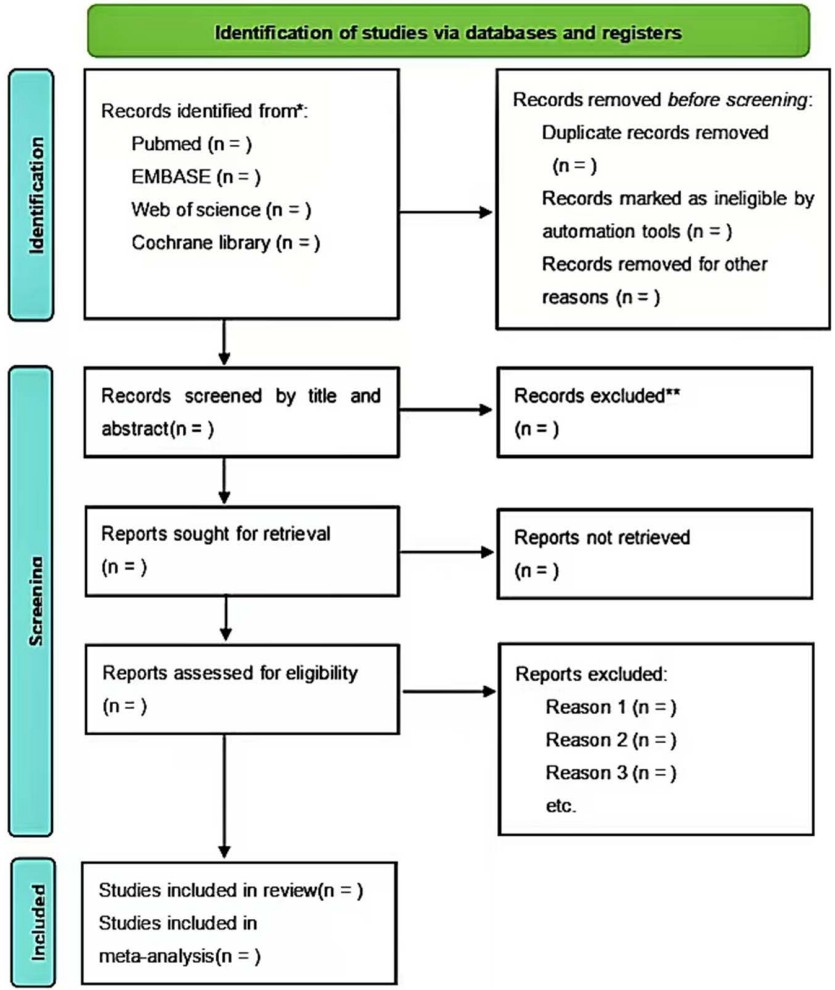

**Fig 1. PRISMA flow diagram of systematic reviews and meta-analyses.**

## Assessment of reporting bias

Two independent reviewers(WFW and XLQ) will use the Risk of Bias due to Missing Evidence tool to assess the risk of bias due to missing evidence [21], and if there is a disagreement, it will be resolved through discussion or consultation with the third reviewer (XCJ). We will assess publication bias with Egger's test (p < 0.05 for bias). When at least 10 studies are included, we will plot funnel plots.If bias is detected, the trim-and-fill method will be used to evaluate its impact on the results.

## Quality of the evidence

The certainty of the evidence will be evaluated using the Grading of Recommendations Assessment, Development, and Evaluation (GRADE) approach by two researchers(WFW and XLQ) and disagreements will be resolved by consultation with the third investigator (XCJ). The scoring method considers the limitations of the study, including the risk of study bias, directness, consistency, precision and publication bias, in order to evaluate the certainty of the combined effect size [22].

## Discussion

Pain in burn victims has always been a serious problem. Wound dressing changes are the most common surgery performed by burn patients. Pain is a problem that can't be ignored in the process of dressing change, which is directly related to the recovery process, quality of life and psychological state of patients. Chronic neuralgia has also been plaguing burn patients, and its causes are complex, including direct nerve damage in burn patients, and long-term pain stimulation can also cause central sensitivity. Opioids are mainstream drugs, and long-term use will reduce the pain threshold and cause hyperalgesia, which will also aggravate the development of chronic neuralgia. Opioid reduction, multimodal analgesia is a good option. Nitrous oxide appears to be a highly safe, fast-acting, and easy-to-operate analgesic for pain management procedures.

A recent systematic review and meta-analysis provided a pooled analysis of the efficacy of nitrous oxide in wound care in adults, including some burn patients [23]. However, this study only performed subgroup analyses based on wound type. Currently, there are no studies evaluating the efficacy and safety of this treatment in burn patients.

The protocol outlines a systematic review and meta-analysis designed to assess the effectiveness and adverse effects of nitrous oxide for pain management in burn dressing change patients. Based on the PRISMA-P checklist and following the methodological guidance provided in the Cochrane Handbook of Systematic Reviews of Interventions, we assessed the methodological quality of existing studies using the GRADE method to comprehensively assess evidence from multiple study designs. These findings will validate the efficacy and safety of nitrous oxide, fill gaps in evidence synthesis, and provide evidence-based treatment options for operational pain relief in burn patients.

## Conclusion

If the results of this meta-analysis show that nitrous oxide has a good analgesic effect on the process of burn dressing changes and does not cause serious side effects, it will promote the clinical promotion and application of nitrous oxide.

### Limitations

- Nitrous oxide concentrations used in different studies may differ.This also affects heterogeneity if the number of studies does not allow for subgroup analysis based on gs concentration.

- The timing of patient inhalation of intervention gases and different time points of measurement results may also constitute a potential source of clinical heterogeneity.

## Supporting information

**S1 Checklist. PRISMA-P (Preferred Reporting Items for Systematic Review and Meta-Analysis Protocols) 2015 checklist completed for the study protocol.**
(DOCX)

## Acknowledgments

We sincerely thank all those involved in the study and all investigators and clinical staff involved in the trial for their efforts.

## Author contributions

**Conceptualization:** Weifeng Wang, Chenxi Sun.

**Methodology:** Xianli Meng, Xiaochen Jiang, Xueling Qiu.

**Project administration:** Yan Zhao, Wei Gong.

**Software:** Fan Sun.

**Supervision:** Lu Tang.

**Writing – original draft:** Weifeng Wang, Yuchen Wang.

**Writing – review & editing:** Weifeng Wang, Wenjuan Cao.

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
