## [Decision Letter · Decision Letter 0]

23 Jan 2026

Dear Dr.  Tang,

Thank you for submitting your manuscript to PLOS ONE. After careful consideration, we feel that it has merit but does not fully meet PLOS ONE’s publication criteria as it currently stands. Therefore, we invite you to submit a revised version of the manuscript that addresses the points raised during the review process.

We look forward to receiving your revised manuscript.

Kind regards,

Gisele Viana de Oliveira

Academic Editor

PLOS One

Journal Requirements:

3. We notice that your supplementary figures are uploaded with the file type 'Figure'. Please amend the file type to 'Supporting Information'. Please ensure that each Supporting Information file has a legend listed in the manuscript after the references list.

Reviewer's Responses to Questions

**Comments to the Author**

1. Does the manuscript provide a valid rationale for the proposed study, with clearly identified and justified research questions?

Reviewer #1: Yes

2. Is the protocol technically sound and planned in a manner that will lead to a meaningful outcome and allow testing the stated hypotheses?

Reviewer #1: Yes

3. Is the methodology feasible and described in sufficient detail to allow the work to be replicable?

Reviewer #1: No

4. Have the authors described where all data underlying the findings will be made available when the study is complete?

Reviewer #1: Yes

5. Is the manuscript presented in an intelligible fashion and written in standard English?

Reviewer #1: Yes

You may also provide optional suggestions and comments to authors that they might find helpful in planning their study.

Reviewer #1: Major Comments

1. Inconsistency in Study Design Eligibility

The protocol states that RCTs will be considered but also indicates that observational studies may be included based on article content.

Clarify one of the following:

• Restrict inclusion strictly to RCTs, or

• Explicitly define observational study types, specify a separate risk-of-bias tool

2. Participant Definition and Age Restriction

The protocol limits participants to patients >18 years old with a first burn, yet:

• Many burn dressing studies include pediatric populations.

• No justification is provided for excluding children.

• The term “first burn” is unclear and not standard.

• Clarify whether “first burn” refers to a first-degree burn or first burn episode.

• Provide a rationale for adult-only inclusion or consider pediatric subgroup analysis.

3. Outcome Definition and Measurement Issues

The primary outcome is defined as change in pain scores before and after intervention, yet:

• Different pain scales (VAS, NRS, FLACC) are proposed.

• FLACC is primarily pediatric, conflicting with adult-only inclusion.

• Change scores vs post-intervention scores may introduce heterogeneity.

• Clarify how different pain scales will be standardized (e.g., standardized mean difference).

4. Comparator Definition Is Too Broad

The comparator includes placebo or other analgesic methods, which may include:

Opioids

• Ketamine

• Regional techniques

• Non-pharmacological interventions

Pooling these comparators risks clinical heterogeneity.

• Predefine comparator categories.

• State whether separate meta-analyses or subgroup analyses will be performed by comparator type.

5. Search Strategy and Language Claims

The protocol states no language restrictions, yet:

• Search terms are predominantly English.

• No plan for translation of non-English studies is described.

Describe how non-English articles will be screened, translated, and assessed to ensure the claim of no language restriction is credible.

6. Statistical Analysis Plan Needs Refinement

The plan specifies MD and RR but does not address:

• Use of standardized mean difference (SMD) when different pain scales are used.

• Add explicit plans for crossover designs.

• Clarify whether repeated measures will be adjusted or excluded.

• Specify SMD use where appropriate.

7. Publication Bias Assessment Thresholds

Egger’s test is proposed with p < 0.1, which is unconventional.

Justify the use of p < 0.1 or align with conventional thresholds (p < 0.05), particularly given expected small study numbers.

Minor Comments

1. Grammar and Style

•Numerous minor grammatical errors and spacing issues (e.g., missing spaces after periods).

•Recommend professional language editing.

2. Terminology

•Use consistent terms: dressing change vs dressing replacement vs “wound care.

**Do you want your identity to be public for this peer review?** For information about this choice, including consent withdrawal, please see our For information about this choice, including consent withdrawal, please see our Privacy Policy .

Reviewer #1: No

---

## [Author Response · Author response to Decision Letter 1]

28 Jan 2026

To editor:

Thank you very much for your help! We have made corrections as per your request.Sorry I didn't specifically indicate that the uploaded image is not supplementary figures, it was uploaded as an image file, the legend is on line 209.Please feel free to contact me if you still have questions. I wish you all the best!

Reviewer 1:

Thank you very much for your careful review!We have made the following corrections to your problem.

1. After careful consideration, we decided to strictly limit the inclusion of RCTs, which can improve the quality of evidence.

2. The reason why we only include adults is that the pain scores of children's burns are mainly assessed by medical staff through facial expressions and leg movements (FLACC scale), while adults mostly use VAS or NRS scales for self-evaluation. Different evaluation subjects may lead to heterogeneity. But you're right, it's true that burns in children are more common, and we can do subgroup analysis of children, so we made a modification in the text to include patients of all ages and do subgroup analysis according to age. The word "first time" was also clarified.

3. Different scales cannot be directly compared. We will use standardized mean differences for meta-analysis.

4. We did overlook this point, and we have added content to the "Comparators" section and the "Subgroup analysis" section.

5. You are very rigorous and the issues you point out are very important. We have added an evaluation method for non-English articles in the Screening section.

6. We corrected this error in the "Data analysis" section. Different scales do use standardized mean differences. We have added the solution to the problem of multiple measurements in the "Data extraction and records" section.

7. We agree that p < 0.05 is a more conventional threshold for hypothesis testing, and this error has been corrected.

I wish you all the best!

---

## [Decision Letter · Decision Letter 1]

16 Feb 2026

Dear Dr. Tang

Thank you for submitting your manuscript to PLOS ONE. After careful consideration, we feel that it has merit but does not fully meet PLOS ONE’s publication criteria as it currently stands. Therefore, we invite you to submit a revised version of the manuscript that addresses the points raised during the review process.

publication criteria  . .

We look forward to receiving your revised manuscript.

Kind regards,

Gisele Viana de Oliveira

Academic Editor

PLOS One

Journal Requirements:

Reviewers' comments:

Reviewer's Responses to Questions

**Comments to the Author**

1. Does the manuscript provide a valid rationale for the proposed study, with clearly identified and justified research questions?

Reviewer #2: Yes

2. Is the protocol technically sound and planned in a manner that will lead to a meaningful outcome and allow testing the stated hypotheses?

Reviewer #2: Yes

3. Is the methodology feasible and described in sufficient detail to allow the work to be replicable?

Reviewer #2: Yes

4. Have the authors described where all data underlying the findings will be made available when the study is complete?

Reviewer #2: Yes

5. Is the manuscript presented in an intelligible fashion and written in standard English?

Reviewer #2: Yes

You may also provide optional suggestions and comments to authors that they might find helpful in planning their study.

Reviewer #2: This paper is the good protocol paper and can be used as an alternative method for burn patients. It is well organized and easy to be understand. However, there is some issue need to be address by the authors.

Methods:

1. Kindly please list down as many or much as possible for another synonym name or keywords for every word used or apply during the search strategies as other authors might use different word, but it refers to the similar meaning or words.

2. As this review will use or focus the clinical trial study design, maybe authors might consider putting or write study design during the search strategies.

3. Kindly please standardized the search strategies as well. For example, for the Embase database in table 2, the EXP is writing in the 1 column. The application of EXP is similar with the MESH term. Kindly, please use MESH for the Nitrous Oxide in the PubMed (Table 1) and Cochrane (Table 4). Similar comment is applied for other keywords that search with EXP in the Embase database (Table 2).

4. The gold standard of performing any systematic review and meta-analysis is by following the Cochrane database guideline. Cochrane encourages other people to perform the search strategies by searching the words one per one, not pull all the keywords in the one column. For example, in Table 1, author write Search: "Nitrous Oxide"[Title/Abstract]OR"Laughing Gas"[Title/Abstract]. Author should search the word one by one and later combined it. The correct way to do it is Search: "Nitrous Oxide"[Title/Abstract], #1. Then do another searching, Search: "Laughing Gas"[Title/Abstract], #2. Then combine the words using OR. #1 OR #2.

5. Under the study type section, for statement follow the PICOS framework (line no 169), it will be better if the author can write: P refers to “Population”, I refer to “Intervention etc. The details on the PICOS will be explain in the next section.

6. Suggest changing the ‘word so’ in line 174 to, ‘therefore’. Same comments for other word that used ‘so’.

7. Under the outcome measures in line 183, kindly please spell out the full name of the tools used: VAS, NRS, or FLACC score.

8. How do authors contact the corresponding author if the authors unable to obtain the data needed such as mean or SD? Is it via email? How many times will the authors follow-up with the corresponding authors to obtain the data. If

9. In line 202, kindly please change the capital C at the control to the lowercase c.

Others:

1. Kindly, please provide the PRISMA Checklist for systematic review.

**Do you want your identity to be public for this peer review?** For information about this choice, including consent withdrawal, please see our For information about this choice, including consent withdrawal, please see our Privacy Policy .

Reviewer #2: No

---

## [Author Response · Author response to Decision Letter 2]

20 Feb 2026

To editor:

Thank you very much for your help!Please feel free to contact me if you still have questions. I wish you all the best!

Reviewer 1:

Thank you very much for your careful review!We have made the following corrections to your problem.

1. We have listed more synonyms and revised the search strategy.

2. We did not list the study design in the search strategy to prevent the omission of eligible studies. We will limit the type of study during manual screening.

3. We have optimized the search strategy according to your requirements in the table.

4. We have modified the retrieval strategy according to your example.

According to your comments on articles 5, 6, 7 and 9, we have revised the text.

8. We have made modifications in the Data extraction and records section of the article.

PRISMA Checklist We have uploaded as other files.

I wish you all the best!

---

## [Decision Letter · Decision Letter 2]

5 Apr 2026

A systematic review protocol:efficacy and safety of nitrous oxide in analgesia in burn patients with dressing change

PONE-D-25-49811R2

Dear Dr. Tang

We’re pleased to inform you that your manuscript has been judged scientifically suitable for publication and will be formally accepted for publication once it meets all outstanding technical requirements.

Kind regards,

Gisele Viana de Oliveira

Academic Editor

PLOS One

Additional Editor Comments (optional):

Reviewers' comments:

Reviewer's Responses to Questions

**Comments to the Author**

1. Does the manuscript provide a valid rationale for the proposed study, with clearly identified and justified research questions?

Reviewer #2: Yes

2. Is the protocol technically sound and planned in a manner that will lead to a meaningful outcome and allow testing the stated hypotheses?

Reviewer #2: Yes

3. Is the methodology feasible and described in sufficient detail to allow the work to be replicable?

Reviewer #2: Yes

4. Have the authors described where all data underlying the findings will be made available when the study is complete?

Reviewer #2: Yes

5. Is the manuscript presented in an intelligible fashion and written in standard English?

Reviewer #2: Yes

You may also provide optional suggestions and comments to authors that they might find helpful in planning their study.

Reviewer #2: Dear Author,

Thank you for answering the comments very well.

However, there is a comment I would like to address in the search strategies. I had attached the correct way to do the searching in the PubMed. Kindly, please made amendment for your search strategies mainly in the PubMed. It is a very good practice that author make a Boolean operator (OR or AND) after the author list down all the synonym by the component. For example, if the study focused on the children: P (population), it should be written as this:

#1: ("Child*"):ti,ab,kw

#2: ("Neonate*"):ti,ab,kw

#3: ("Toddler*"):ti,ab,kw

#4: #1 OR #2 OR #3 OR #4

Thanks

**Do you want your identity to be public for this peer review?** For information about this choice, including consent withdrawal, please see our For information about this choice, including consent withdrawal, please see our Privacy Policy .

Reviewer #2: No

---

## [Editor Report · Acceptance letter]

PONE-D-25-49811R2

PLOS One

Dear Dr. Tang,

I'm pleased to inform you that your manuscript has been deemed suitable for publication in PLOS One. Congratulations! Your manuscript is now being handed over to our production team.

Kind regards,

on behalf of

Dr. Gisele Viana de Oliveira

Academic Editor

PLOS One